# Detection of gene cis-regulatory element perturbations in single-cell transcriptomes

Grace Hui Ting Yeo[1,2], Oscar Juez[3], Qing Chen[3], Budhaditya Banerjee[3], Lendy Chu[3], Max W. Shen[1,2], May Sabry[4], Ive Logister[4], Richard I. Sherwood[3,4]*, David K. Gifford[1]*

1 Computer Science and Artificial Intelligence Laboratory, Massachusetts Institute of Technology, Cambridge, Massachusetts, United States of America, 2 Computational and Systems Biology Program, Massachusetts Institute of Technology, Cambridge, Massachusetts, United States of America, 3 Division of Genetics, Department of Medicine, Brigham and Women's Hospital and Harvard Medical School, Boston, Massachusetts, United States of America, 4 Hubrecht Institute, Utrecht, the Netherlands

* rsherwood@rics.bwh.harvard.edu (RIS); gifford@mit.edu (DKG)

## Abstract

We introduce poly-adenine CRISPR gRNA-based single-cell RNA-sequencing (pAC-Seq), a method that enables the direct observation of guide RNAs (gRNAs) in scRNA-seq. We use pAC-Seq to assess the phenotypic consequences of CRISPR/Cas9 based alterations of gene cis-regulatory regions. We show that pAC-Seq is able to detect cis-regulatory-induced alteration of target gene expression even when biallelic loss of target gene expression occurs in only ~5% of cells. This low rate of biallelic loss significantly increases the number of cells required to detect the consequences of changes to the regulatory genome, but can be ameliorated by transcript-targeted sequencing. Based on our experimental results we model the power to detect regulatory genome induced transcriptomic effects based on the rate of mono/biallelic loss, baseline gene expression, and the number of cells per target gRNA.

**Data Availability Statement:** Raw and processed single-cell RNA-sequencing data are available from GEO (accession number GSE117053). All other

## Author summary

Predicting how mutations in non-coding regions impact gene expression is an important step towards understanding how non-coding variants contribute to human variation and disease. CRISPR/Cas9 enables programmable and scalable alteration of genomic regions of interest. More recently, CRISPR/Cas9 based screens have been combined with single-cell RNA-sequencing (scRNA-seq) technology to provide cellular transcriptomes as screen readouts. However, existing studies coupling CRISPR screens with scRNA-seq have focused on perturbations to open reading frames (ORFs). We develop an assay for directly observing CRISPR/Cas9 guide RNAs (gRNAs) in scRNA-seq, and apply it to monitoring gene expression changes induced by perturbations to the regulatory genome. We find that in contrast to ORF targeting, altering regulatory regions rarely results in total knock-out of the targeted gene. We analyze how the power to detect the consequences of changes to the regulatory genome depends on factors such as number of cells

relevant data are available in the manuscript and its Supporting Information files.

**Funding:** We acknowledge funding from NIH grants 1R01HG008363 (D.K.G.), 1R01HG008754 (D.K.G.), and 1K01DK101684-01 (R.I.S.); the Human Frontier Science Program, Netherlands Organisation for Scientific Research, Brigham Research Institute, and Harvard Stem Cell Institute (R.I.S.); and the Agency for Science, Technology and Research Graduate Academy (G.H.T.Y.). The funders had no role in study design, data collection and analysis, decision to publish, or preparation of the manuscript.

**Competing interests:** I have read the journal's policy and the authors of this manuscript have the following competing interests: David K. Gifford is a founder of Think Therapeutics, Inc.

receiving a gRNA, the extent of a knock-out, and the baseline gene expression in the control.

## Introduction

A large fraction of genetic variation associated with human disease lies in the non-coding region of the human genome [1]. Predicting how non-coding mutations impact phenotype however remains a difficult challenge. While epigenetic atlases have collated large datasets of epigenetic signatures associated with regulatory elements, they have not been able to precisely identify which elements are functional, or their regulatory effects on gene expression. High-throughput functional assays are hence important to directly interrogate the non-coding genome [2,3]. Popular approaches include MPRA (massively parallel reporter assay) and STARR-seq (self-transcribing active regulatory region sequencing), both of which utilize high-throughput sequencing of large reporter plasmid libraries to assay either synthetic oligonucleotides or DNA fragments derived from genomic sources. However, one limitation is that plasmid-based reporter assays do not interrogate sequences within their chromatin and genomic context [4–6].

In contrast, CRISPR/Cas9 technology enables programmable genome editing within the endogenous context by using guide RNAs (gRNAs) to target genomic regions of choice [7–10]. Genome-scale gene open reading frame knock-out screens have provided key links between gene function and disease phenotypes such as cancer drug resistance [11–13]. Tiled mutation of non-coding regions has also pinpointed regulatory regions crucial in controlling nearby gene expression [14–17]. CRISPR/Cas9 genome mutation screens have for the most part relied on low-dimensional readouts such as cell survival or reporter gene expression that provide a limited picture of the consequences of genome mutation. More recently, these screens have been paired with droplet-based single-cell RNA-sequencing (scRNA-seq) to enable high-throughput, high-dimensional readouts of CRISPR/Cas9-induced gene mutations in the form of cellular transcriptomes [18–21]. Alternatively, a mid-throughput option is targeted scRNA-seq panels, which can be used to focus the sequencing budget on only the genes of interest and are now commercially available via 10X [22,23]. Regardless, these pioneering single-cell studies have focused on targeting gene coding regions, and the potential of coupling CRISPR/Cas9 with scRNA-seq for functional annotation of non-coding regions remains largely unexplored. A few exceptions include Xie et al. and Gasperini et al. [24,25], which sought to map enhancer-gene interactions using CRISPRi and scRNA-seq. However, CRISPRi is limited in resolution as it typically makes use of a KRAB repressor domain tethered to a nuclease-inactivated Cas9 to induce heterochromatin over 1-2kb. Furthermore, the KRAB repressor domain may not accurately perturb enhancer function at distal sites [26]. This is in contrast to CRISPR mutational screens, which depend on error-prone repair following CRISPR/Cas9-induced double-stranded breaks. These screens have previously been shown to allow dissection of non-coding regions at single-base resolution [14].

Existing CRISPR screens that are observed by scRNA-seq do not optimally solve the technical challenge of observing which CRISPR/Cas9 guide RNA (gRNA) is in a given cell. Because current scalable scRNA-seq techniques rely on detecting poly-adenylated transcripts and gRNAs are not natively poly-adenylated, existing approaches infer the presence of gRNAs by the observation of separately transcribed polyadenylated barcode transcripts that are associated with gRNAs. This approach complicates the design of gRNA vectors that can be used in scRNA-seq, requires gRNA construct sequencing to link barcodes with their corresponding

gRNAs, and is susceptible to high rates of mislabeling of gRNAs due to lentiviral recombination, which can result in barcode swapping rates exceeding 50% [27–29]. Additionally, linked barcodes do not allow for screens based on direct administration of *in vitro* transcribed gRNA such as commonly used ribonucleoprotein (RNP) complexes [30,31]. CROP-seq has proposed to solve this using a construct where the gRNA cassette is duplicated during viral integration in such a way that an RNA polymerase III transcript for genome editing and an RNA polymerase II transcript for detection by scRNA-seq are both expressed [20]. Others have proposed introducing guide-specific primers [22].

We have developed a method of directly observing gRNA presence in cells called poly-adenine CRISPR gRNA-based single-cell RNA-sequencing (pAC-Seq), that appends a poly-adenine tract at the 3' end of gRNA sequences. We have found that adding a poly-adenine tract maintains the full mutagenic activity of gRNAs. These gRNAs are robustly detectable in conventional scRNA-seq experiments and thus enable the confident association of gRNA induced edits to their consequential changes in a cell's transcriptome. We employ pAC-Seq to monitor changes in gene expression induced by CRISPR/Cas9-induced non-coding mutations in cis-regulatory regions previously shown to disrupt gene expression. We show that pAC-Seq is able to detect cis-regulatory-induced loss of target gene expression even when biallelic loss of target gene expression occurs in only ~5% of cells. However, we observe that even when using cis-regulatory region targeting gRNAs pre-selected to be maximally disruptive to gene expression, ~20–30% of cells have some loss of target expression, while ~5% of cells have biallelic loss of target gene expression. Using a simulation-based power analysis we show how the power to detect transcriptomic effects depends on several factors including the rate of mono/biallelic loss, base gene expression, and the number of cells per target gRNA.

## Results

### Adding a 3' poly-adenine tract to gRNAs enables robust detection in oligo (dT)-primed reverse transcription without compromising mutagenic activity

The Streptococcus Pyogenes Cas9 (spCas9) single guide RNA (gRNA) sequence has been shown to be amenable to a number of sequence alterations including 3' sequence insertions. To test whether adding a 3' poly-adenine tract to the gRNA is compatible with Cas9 mutagenic activity, we designed a spCas9 plasmid gRNA sequence with 25 adenines between the end of the gRNA sequence and the transcriptional terminator sequence (Fig 1A). We cloned this 25-adenine gRNA hairpin (25A-gRNA) into our standard gRNA plasmid backbone which includes a U6 promoter and Hygromycin resistance cassette flanked by Tol2 transposon sites. We then cloned three GFP-targeting spacers into either the wildtype gRNA or the 25A-gRNA plasmids. We co-introduced each gRNA along with a plasmid containing CBh promoter-driven Cas9 and Blasticidin resistance flanked by Tol2 sites as well as a Tol2 transposase plasmid into Zfp42-GFP knock-in mouse embryonic stem cells (mESCs) [14]. After stable selection of Blasticidin and Hygromycin dually resistant cells, we found that with all 3 GFP-targeting gRNAs in the wildtype and 25A-gRNA backbones, >99% of cells lost GFP expression (Fig 1B). We additionally tested 9 distinct genome-targeting spacers in the 25A-gRNA format, achieving near complete mutation in all cases (S1 Fig).

Having confirmed mutagenic activity of 25A-gRNAs, we next assessed the relative abundance of gRNA in oligo(dT)-primed reverse transcription reactions since all existing high-throughput scRNA-seq techniques rely on such priming. We note that wt-gRNAs are not poly-adenylated by cells because they are transcribed by a polymerase III promoter. We found that 25A-gRNAs are 13.8X more abundant than standard gRNAs in oligo(dT)-primed RT-

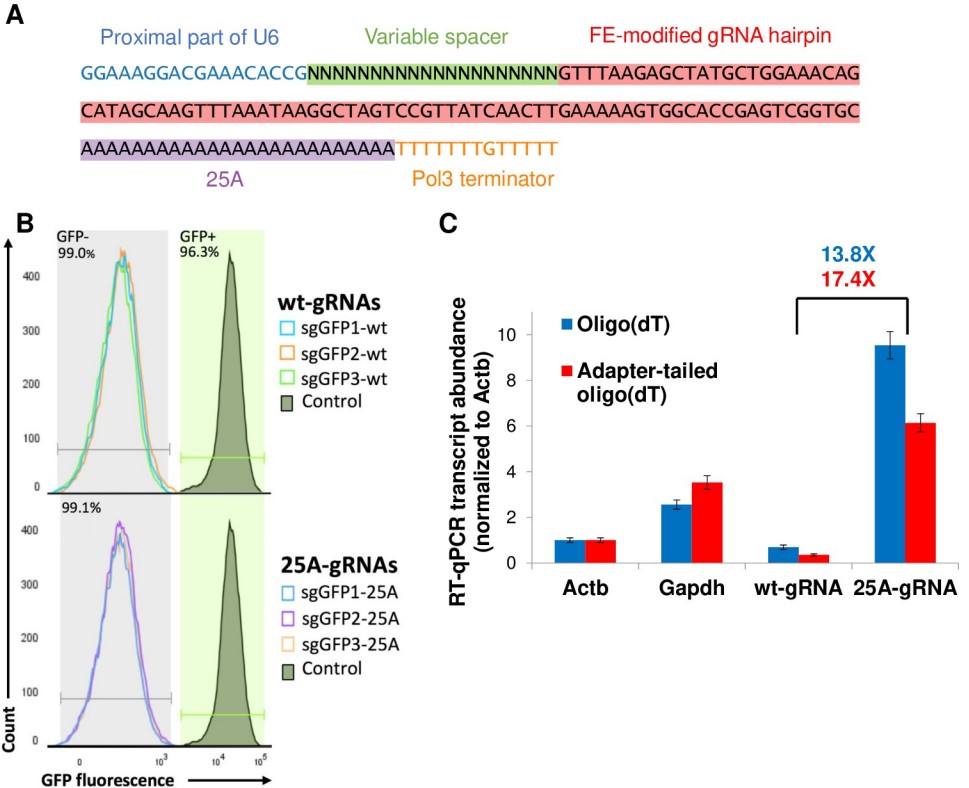

**Fig 1. Polyadenylated guide RNAs retain activity and are robustly detectable in dT-primed reverse transcription.**
(A) Schematic of poly adenine-tailed gRNA (25A-gRNA) structure. Highlighted regions constitute the gRNA transcript, while unhighlighted regions flanking the gRNA transcript are part of the plasmid but not the transcript. The FE-modified gRNA hairpin is as in Chen et al. [32] (B) Flow cytometry plot comparing effectiveness of three wildtype gRNAs (wt-gRNAs) and 25A-gRNAs at knockout of GFP in *Zfp42*^GFP mESCs after stable transfection along with Cas9. Horizontal segments indicate gates for GFP+ and GFP- cells (C) RT-qPCR comparison of wt-gRNA vs 25A-gRNA abundance using two different forms of dT priming, normalized to Actb expression.

qPCR and 17.4X more abundant when using a sequencing adapter-tailed oligo(dT) primer as is used in scRNA-seq (Fig 1C). 25A-gRNAs but not standard gRNAs are more abundant than transcripts of the housekeeping genes Actb and Gapdh which are identified in nearly all cells in droplet-based scRNA-seq [33], suggesting that 25A-gRNAs should be robustly identifiable in scRNA-seq data.

## Mutations at regulatory regions produce a diverse array of genotypic and phenotypic outcomes

We have previously used a tiled CRISPR/Cas9 screening technique called MERA to identify non-coding regulatory regions required for the expression of a gene [14]. One finding from this work was that CRISPR/Cas9-induced mutations caused by a single cis-regulatory region-targeting gRNA (cis-gRNA) cause a wide spectrum of repair genotypes, only a small subset of which lead to loss of expression of a fluorescently tagged gene. We thus chose to explore the heterogeneous outcomes of cis-gRNA targeting using gRNAs previously observed to cause loss of gene expression as a proof-of-principle system to evaluate the performance of 25A-gRNAs in scRNA-seq.

To identify cis-gRNAs with variable phenotypic effect on target gene expression, we performed MERA screening in a 224 kb region surrounding the Msh2 tumor suppressor gene in

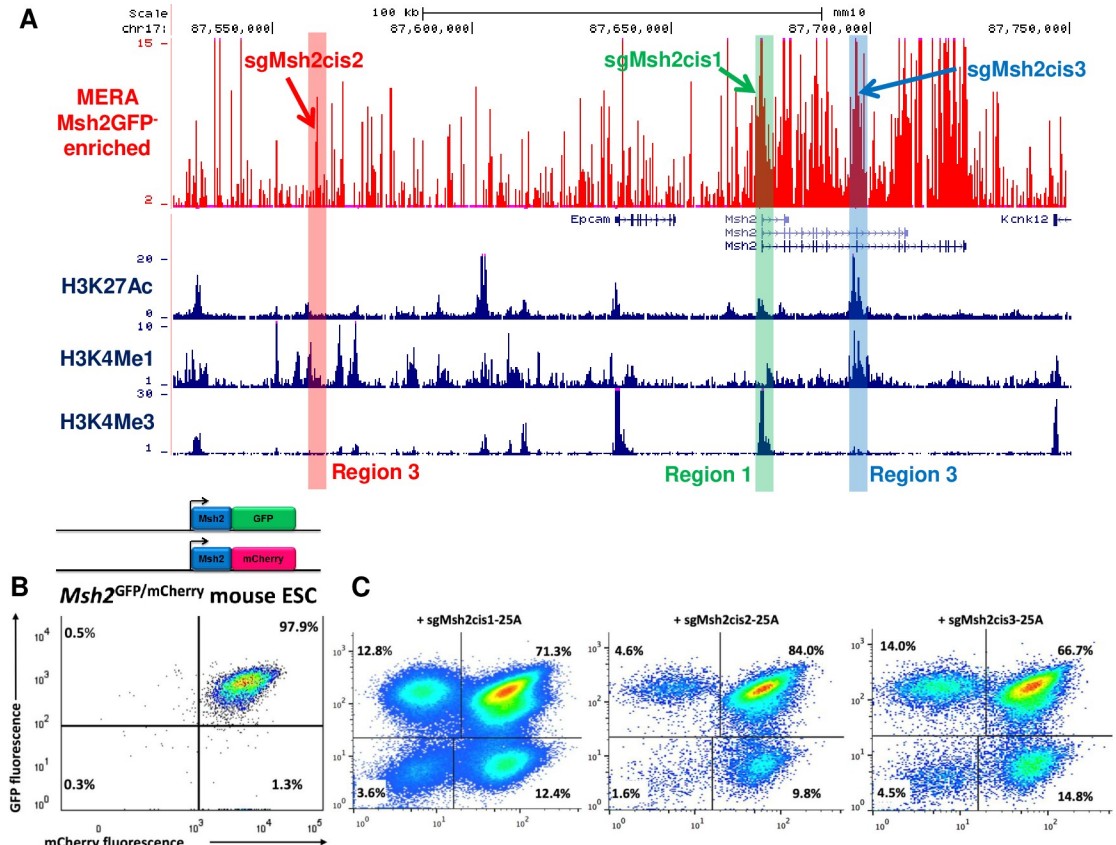

**Fig 2. Mutations at Msh2 regulatory regions infrequently induce biallelic loss of expression.** (A) UCSC browser track showing MERA screen ratio of gRNA abundance in Msh2GFP⁻ vs bulk pools for ~12,000 gRNAs tiled across the Msh2 cis-regulatory region in mESCs. ENCODE histone modification tracks are shown underneath plot, and gRNAs chosen for follow-up are highlighted. (B) Flow cytometry plot showing $Msh2^{GFP/mCherry}$ mESC line before CRISPR/Cas9 targeting. (C) Flow cytometry plots of $Msh2^{GFP/mCherry}$ mESCs after targeting with three cis-gRNAs, showing infrequent biallelic loss of expression.

mESCs. Monoallelic inactivation of the human homologue of Msh2 causes Lynch Syndrome, a hereditary cancer syndrome characterized by high frequencies of colon and other cancers [34,35]. Msh2 biallelic knockout mice are viable, although they are more susceptible to cancer [36,37]. Msh2 is constitutively expressed in mESCs and their derivatives, yet the phenotypic effects of its loss in ESCs is poorly understood. We produced Msh2-GFP fusion mESCs and performed two replicates of MERA screening. We identified a number of required regulatory regions including the Msh2 promoter (Region 1), an upstream region without classic active histone marks (Region 2), and an intronic enhancer region (Region 3) (Fig 2A and S1 Table) which fits into a paradigm of unmarked regulatory elements (UREs) we have previously identified [14]. To confirm that these regions do in fact cause loss of Msh2 expression, we performed paired gRNA deletions of these three regions, finding significant enrichment of GFP loss (one-sided t-test p-value < 0.05) upon deletion of these regions as compared to deletion of control regions (S2 Fig).

To delve deeper into the heterogeneity of loss-of-expression induced by cis-gRNA targeting, we constructed mESCs in which each allele of Msh2 or the highly expressed but non-essential mESC-specific Tdgf1 gene is marked by a distinct fluorescent protein. We constructed Msh2-GFP/Msh2-mCherry and Tdgf1-GFP/Tdgf1-mCherry mESCs using a CRISPR/Cas9 HDR protocol that requires short homology arms [38] (Figs 2B and S3). We then used

these lines to assess the allelic distribution of cis-gRNA-induced loss of expression. For Msh2, we used one gRNA each from Regions 1–3 that induced significant GFP loss in the MERA screen (hereafter sgMsh2cis1-3, Fig 2A). For Tdgf1, we chose one gRNA each from three distinct regions found to induce significant GFP loss in a previously published MERA screen, including the promoter, an intronic enhancer, and an upstream enhancer (hereafter sgTdgfcis1-3, S3 Fig). We performed Tol2-mediated Cas9 and gRNA delivery for all six gRNAs targeting GFP+mCherry+ lines followed by dual selection to ensure that all cells have integrated Cas9 and gRNA. We found that 66–84% of cells remained GFP+mCherry+ (Figs 2C and S3), and most of the cells that lost expression did so monoallelically, with only 1.6–4.8% of cells becoming GFP-mCherry-. This low rate of expression loss was not explained by CRISPR/Cas9 mutation heterogeneity, as a large majority of alleles contain indels (S1 Fig). Thus, even when targeting promoter and strong enhancer regions, we found it is rare to induce biallelic loss of expression through cis-gRNA CRISPR/Cas9-induced mutation.

## High-confidence detection of 25A-gRNAs in scRNA-seq datasets

We then proceeded to test the utility of 25A-gRNAs in scRNA-seq using cis-gRNA targeting as our model system. We performed two scRNA-seq experiments, the first on flow cytometrically purified cells in which gRNA identity should be consistently correlated with target gene expression and any downstream phenotypic change and the second on bulk cells such that gRNA presence only rarely induces loss of target gene expression (Fig 3A). We first describe the methodological aspects of these screens and then proceed to the biological analysis.

In each experiment, we mixed 10 cell lines, each dually drug-selected to express Cas9 and a single gRNA species, at equal ratio and grew them as a pool for 48 hours prior to scRNA-seq. We then collected ~10,000 cells in each experiment using the 10X Chromium scRNA-seq platform. To maximize the simultaneous detectability of gRNAs and their phenotypic effects on single cells, we modified several steps of the scRNA-seq library prep. We refer to the entire pipeline of 25A-gRNA-based CRISPR/Cas9 targeting through scRNA-seq analysis as poly-adenine CRISPR gRNA-based single cell RNA-sequencing (pAC-Seq). The standard protocol is used to prepare cDNA where every molecule is encoded with a cell barcode and UMI. We then performing standard transcriptome-wide Illumina sequencing library preparation on a portion of the cDNA and then use the remainder for custom sequence capture using a gRNA hairpin-specific probe. gRNA-enriched cDNA from this sequence capture is then prepared for Illumina sequencing, which can be pooled with transcriptome sequencing or sequenced separately in order to control the number of gRNA reads obtained. In these experiments, we collected $>4^*10^8$ transcriptome reads per dataset and $>1^*10^7$ reads per gRNA dataset.

We assigned gRNAs to cells with a method that detected gRNA presence by expression above a noise level. We had observed that gRNA count fractions were bimodally distributed, with a signal population of cells containing the gRNA, and a noise population where observed counts are as a result of technical errors such as mapping artefacts. Furthermore, the mean count fraction of these populations varied between gRNAs (S4A Fig). We hence chose to model UMI counts for each gRNA independently as arising from a two-component binomial mixture (Fig 3B). A gRNA was defined to be present in a cell if the probability of the observed count arising from the noise component was less than 0.05. We defined the detection rate as the fraction of cells with high quality transcriptomes (see Methods) that could be mapped to one specific gRNA. Detection rates in the flow cytometrically purified and bulk datasets were 75.3% (of 13016 cells, 3208 unassigned) and 91.2% (of 8887 cells, 859 unassigned) respectively (Fig 3C). Previous publications using indirect detection of gRNAs in scRNA-seq through linked transcripts have reported equivalent detection rates (up to 92.2%) [18,19,21]. However,

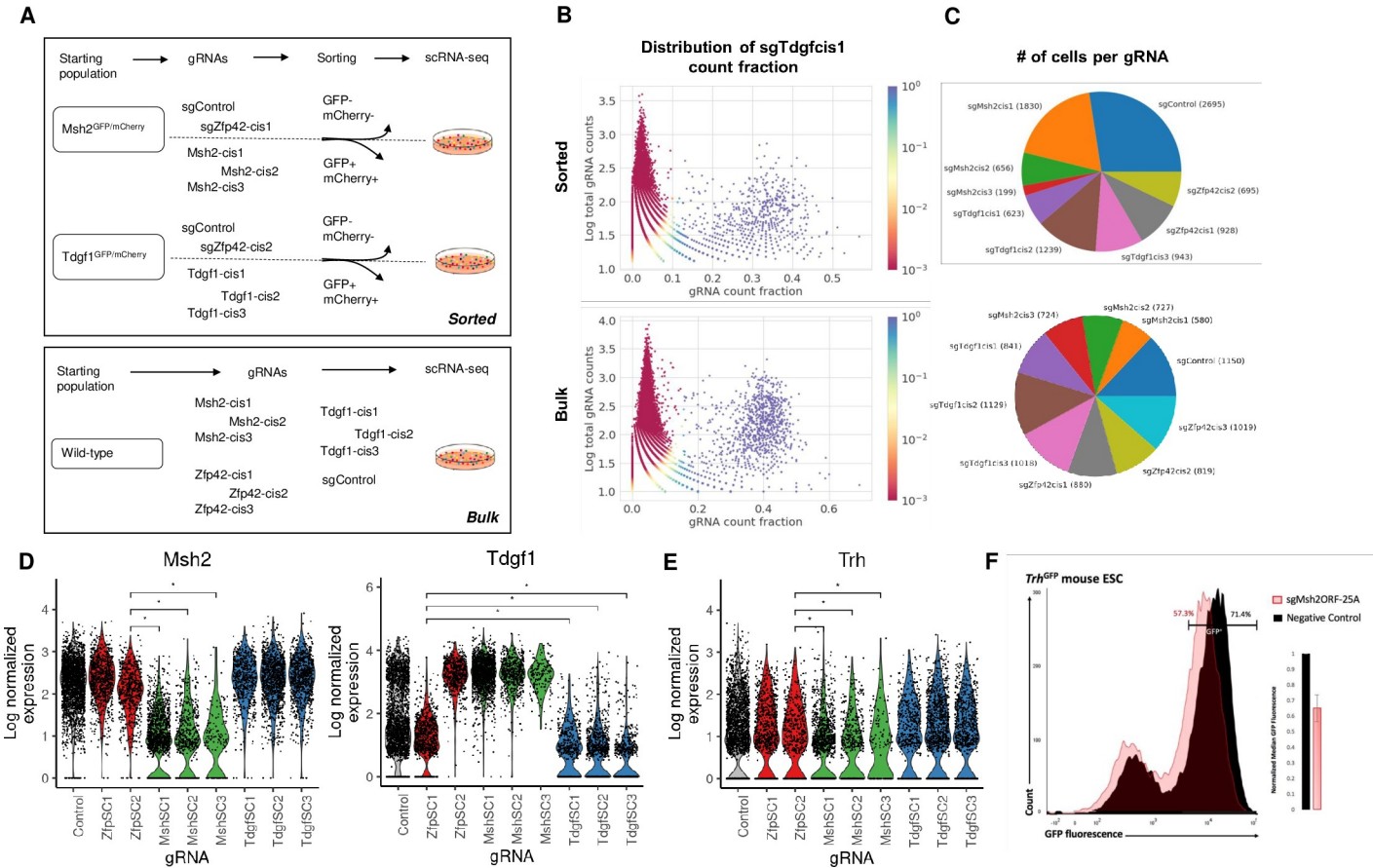

**Fig 3. pAC-Seq faithfully recovers gRNA identity and transcriptomic differences in pooled single cell RNA-seq of pre-sorted populations.** (A). Schematic of two pAC-Seq experimental schemes. In the top experiment, Msh2[GFP/mCherry] and Tdgf1[GFP/mCherry] mESCs were each targeted with three cis-gRNAs, and GFP⁻mCherry⁻ cells were flow cytometrically purified. These same cell lines were each also targeted with two control gRNAs which do not cause loss of target gene expression such that gRNA identity should be highly correlated with target gene expression. The ten populations were pooled for single cell RNA-sequencing (scRNA-seq) on ~10,000 cells. In the bottom experiment, wildtype mESCs were separately targeted with ten gRNAs, one control and three cis-gRNAs each targeting Msh2, Tdgf1, and Zfp42 regulatory regions. The ten populations were pooled for scRNA-seq on ~10,000 cells. (B). Plot showing sgTdgfcis1 gRNA counts as a fraction of all gRNA counts (X-axis) in each cell, plotted against the total gRNA counts in that cell (Y-axis) for the sorted experiment (top) and unsorted experiment (bottom). The coloring represents the probability in the binomial mixture model that sgTdgfcis1 is observed through noise. A clear population can be distinguished in blue ($p<0.05$) in which the cell is called as containing sgTdgfcis1. (C). Pie chart showing the number of cells assigned to each of the ten gRNAs in the sorted (top) and unsorted (bottom) experiments. (D) Violin plots showing log normalized expression of Msh2 (left) and Tdgf1 (right) with transcript-targeted sequencing in the sorted experiment. * denotes statistical significance at adjusted p-value < 0.05. (E) Box plots showing log normalized expression of Trh, which is downregulated in cells that received sgMsh2cis1-3. * denotes statistical significance at adjusted p-value < 0.05 (F) Flow cytometry histogram of Trh-GFP fluorescence in cells with Msh2-ORF targeted knockout (pink) or control gRNA (black), showing that Msh2 knockout subtly downregulates Trh expression. Inset plot shows normalized GFP fluorescence averages in three biological replicate experiments.

these methods are subject to incorrect labeling of gRNAs due to lentiviral recombination, with multiple recent studies reporting barcode swapping rates exceeding 50% [27–29]. Noise in genotypic labels can be addressed via computational imputation approaches, but such approaches can also bias downstream analysis. A previously reported method that also reads out the gRNA directly, CROP-seq, has a far lower detection rate of 45.4%, with a median gRNA readout of 1 UMI/cell [20]. In contrast, we observed a median gRNA readout of 84 and 149 UMIs/cell in the purified and bulk datasets respectively (S2 Table). A newly optimized version of CROP-seq with targeted amplification reported a detection rate of 94% [29].

For the experiments in this work, we separately increased sequencing depth of Msh2, Tdgf1, and Zfp42 transcripts through PCR-based library preparation starting from a portion

of cDNA. We found that focused sequencing of these transcripts dramatically increased the number of UMI-unique reads per gene while maintaining the relative abundance of each transcript per cell (S4B Fig).

## pAC-Seq analysis of cis-gRNA-targeted cells with defined phenotypes

To confirm that pAC-Seq gRNA assignments accurately reflect gRNA-containing cell populations, we performed an experiment in which gRNA identity should correlate strongly and uniformly with target gene expression. We began with either Msh2-GFP/Msh2-mCherry cells or Tdgf1-GFP/Tdg1-mCherry cells, and for cells containing sgMsh2-cis1-3 and sgTdgf1-cis1-3, we performed two rounds of flow cytometric sorting for GFP-mCherry- cells, yielding 85–98% GFP-mCherry- purity (S5 Fig). As a control, we performed Tol2 transfection of each cell line with a control gRNA and a cis-gRNA targeting Zfp42. As expected, these cells were uniformly GFP+mCherry+. We confirmed by RT-qPCR that GFP-mCherry- cells had substantially lower levels of the target transcript than GFP+mCherry+ cells (S6 Fig). In sum, we had 10 cell lines each expressing a single gRNA species in which we predict that expression of the Msh2 and Tdgf1 target genes should be highly correlated with gRNA identity (Fig 3A).

After merging pAC-Seq gRNA assignments with transcriptomes, we found that cells containing sgMsh2-cis1-3 had significantly lower levels of Msh2 transcript than cells containing sgZfp42-cis2 (Wilcoxon rank sum, adjusted $p$-value $< 0.05$) (Fig 3D), showing that pAC-Seq allows accurate assignment of gRNAs to cells. We noticed that cells containing sgZfp42-cis2, which all came from the Msh2-GFP/Msh2-mCherry background, consistently had slightly diminished levels of Msh2 transcript as compared to cells containing sgZfp42-cis1 from the Tdgf1-GFP/Tdgf1-mCherry background (Fig 3D). This difference remained when GFP/mCherry fusion transcripts were taken into account, suggesting that fusion transcripts are expressed at lower levels than wildtype ones, presumably as a result of decreased polymerase processivity or transcript stability. A small percentage of sgMsh2-cis1-3 cells had wild-type levels of Msh2 (Fig 3D), likely a result of imperfect flow cytometric purity. Msh2 is the most downregulated of all genes in Msh2GFP(-)mCherry(-) cells. Differential expression analysis of cells containing sgMsh2-cis1-3 against cells containing sgZfp42-cis2 uncovered a small set of genes with consistent but subtle changes in gene expression including Trh (Fig 3E). We confirmed that Msh2 knockout induces subtle but consistent downregulation of Trh using CRISPR/Cas9-mediated Msh2 coding region mutation in Trh-GFP mESCs (Fig 3F).

We conducted a similar analysis with Tdgf1. We observed that cells containing sgTdgf1-cis1-3 had lower levels of Tdgf1 transcript than cells containing sgZfp42-cis1 (Wilcoxon rank sum, adjusted $p$-value $< 0.05$, Fig 3C). Differential expression analysis did not detect any other consistently differentially expressed genes in sgTdgf1-cis1-3-targeted cells.

## pAC-Seq analysis detects rare cis-gRNA-induced loss of target gene expression

We next applied pAC-Seq to detect rare loss of expression events resulting from cis-gRNA targeting of wildtype mESCs, where we had found that <5% of cells exhibit biallelic loss of expression. We performed stable Tol2 integration of Cas9 and 10 distinct gRNAs: one control, sgMsh2cis1-3, sgTdgf1cis1-3, and sgZfp42cis1-3 (Figs 2A, S3 and S6). Each population was prepared separately such that all cells had exactly one gRNA species, and cells were pooled for pAC-Seq. Sanger sequencing confirmed that these populations had near-complete editing efficiency (S1 Fig).

Cells receiving cis-gRNAs had detectable changes in median expression of their target gene (Wilcoxon rank sum, adjusted $p$-value $< 0.05$) for 6 of 9 gRNAs (Fig 4A). The strength of this

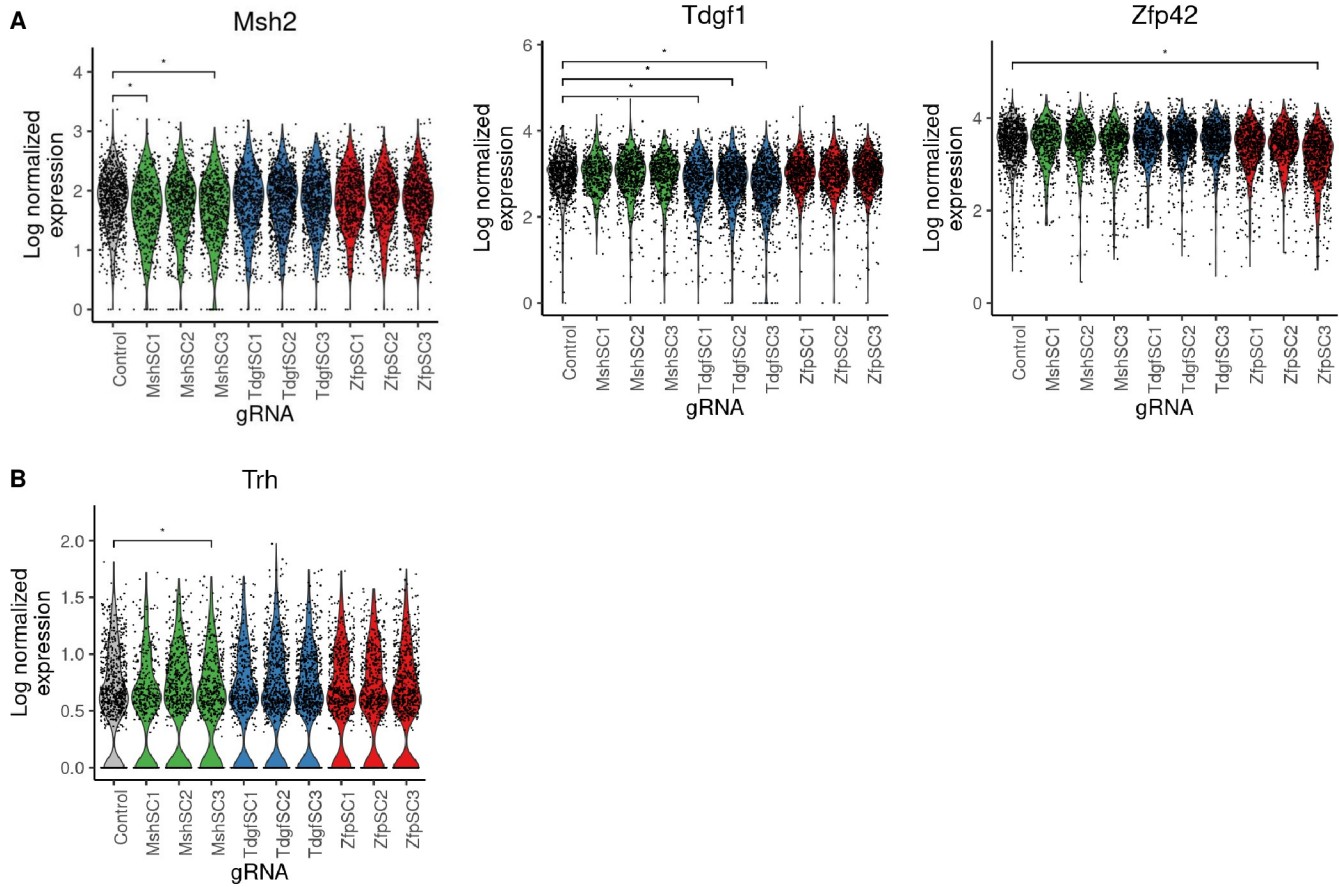

**Fig 4. Detection of rare loss of expression events caused by CRISPR/Cas9 non-coding mutation in wildtype cell pools.** (A) Violin plots showing log normalized expression of Msh2 (left), Tdgf1 (middle), and Zfp42 (right) with transcript-targeted sequencing in the unsorted experiment. * denotes statistical significance at adjusted p-value < 0.05. (B) Violin plots showing log normalized expression of Trh, which is down-regulated in cells where Msh2 was also observed to be significantly down-regulated. * denotes statistical significance at adjusted p-value < 0.05.

loss in target gene expression was consistent with the percent of GFP- cells produced by each gRNA (Figs 2C, S3 and S6). While loss of median target gene expression was observed, the data was insufficient to impute with confidence which cells possess mono-allelic or bi-allelic loss of expression, suggesting a current limit in scRNA-seq resolution (S7B Fig). When gRNA assignments were merged with transcriptomes, graph-based clustering showed no obvious difference in global gene expression when cells were labeled by gRNA (S7A Fig). Trh was found to be differentially expressed in cell populations receiving sgMsh2cis3, in which loss of Msh2 expression was also observed (Fig 4B, Wilcoxon rank sum, adjusted p-value < 0.05). In general, differential expression analysis was underpowered to identify the few subtle downstream effects of Msh2 loss that had been observed in the sorted population.

## A simulated-based power analysis framework guides future experimental design

Our results show that experiments to observe the gene expression consequences of changes to the cis-regulatory genome in single-cells must be carefully designed. To guide future experimental design, we developed a simulation-based analysis framework to predict the ability to observe changes in gene expression. This framework is based on key factors we have observed

to be important in our experiments: (1) the rate of mono/biallelic gene expression loss, (2) baseline gene expression before regulatory pertubation, and (3) the number of cells per target gRNA. We calibrate our framework based on the down-regulation of gene expression we observed in our experimental data.

For our simulation framework, genes were divided into 10 buckets based on their log normalized baseline expression. Then, for each bucket, a gene was randomly drawn. Loss of expression in a treatment population was then simulated for that gene as a mixture of partial and full loss in a random sample of cells from the control population, where full loss corresponds to setting the expression to zero (biallelic loss), and partial loss corresponds to halving expression (monoallelic loss). Differential expression analysis was then conducted via Wilcoxon rank sum with simple Bonferroni correction. The procedure was repeated 100 times for each gene bucket. To simulate conditions in our experiment, we used the partial and full loss fractions for each Msh2 and Tdgf1-targeting gRNA from our FACS data.

This simulation analysis corroborates our experimental observation that the lower the baseline expression of a target gene, the greater the number of treatment cells required to observe significant differential expression. In particular, we had observed in our data that significant down-regulation of Msh2 in wild-type mESCs could only be detected with Msh2 transcripttargeted sequencing and only for cell populations receiving MshSC1 and MshSC3, while significant down-regulation of Tdgf1 was observed across all cell populations receiving Tdgf1-targeting gRNAseven without transcript-targeted sequencing ([Fig 4]). This is supported by our simulations ([Fig 5]), which show that given Msh2's baseline expression without transcript-targeted sequencing, no number of treatment cells enables detection. Furthermore, the analysis also suggests that with Msh2-targeted sequencing, at least 200 cells for MshSC1 and MshSC3, while the number of cells required to detect differential expression of MshSC2 was greater than 900. Given the actual numbers of cells observed, we would then expect to only detect differential expression for MshSC1 and MshSC3, which is as we observed. Similarly, power analysis for Tdgf1-targeting gRNAs show that the design is well-powered to detect Tdgf1 differential expression due to the relatively high baseline expression of Tdgf1 as well as the relatively high number of cells collected for each Tdgf1-targeting gRNA, and despite Tdgf1-targeting gRNAs resulting in lower mono- and bi-allelic loss ([S8 Fig]).

Finally, one way to alleviate the burden of multiple testing would be to perform independent filtering [39] by testing only genes for which we expect to be able to observe significant differential expression. This is because our results have shown that at a given number of treatment cells, we would not expect to be able to observe significant differential expression for genes with mean baseline expression below an experiment specific threshold. Hence, for each bucket, we also considered only testing and hence adjusting for genes with mean baseline expression higher than the gene with the lowest expression in that bucket. If we limit the testing scope to at least the bucket in which the target gene falls, no minimum is detected for MshSC1 and MshSC3, while ~400 cells is required for MshSC2.

The analysis framework can hence be used to predict the number of total cells required to detect differential expression of genes of interest, or if transcript-targeted sequencing will be required to enable detection. The simulation framework is implemented in R (powerpAC), and requires as user input the expected mono/bi-allelic loss rate of the gRNA, as well as the scRNA-seq expression profile of a control population. Then, given the mean log expression of a gene of interest in an unperturbed control, the analysis indicates the p-value that is predicted to be achieved given the number of cells in the experiment when testing for the loss of gene expression. Resulting plots (Figs [5], [S8] and [S9]) from the analysis indicate cell counts required for a desired uncorrected p-value, a p-value Bonferroni corrected for the number of all genes, and a p-value Bonferroni corrected for the number of genes after independent filtering of

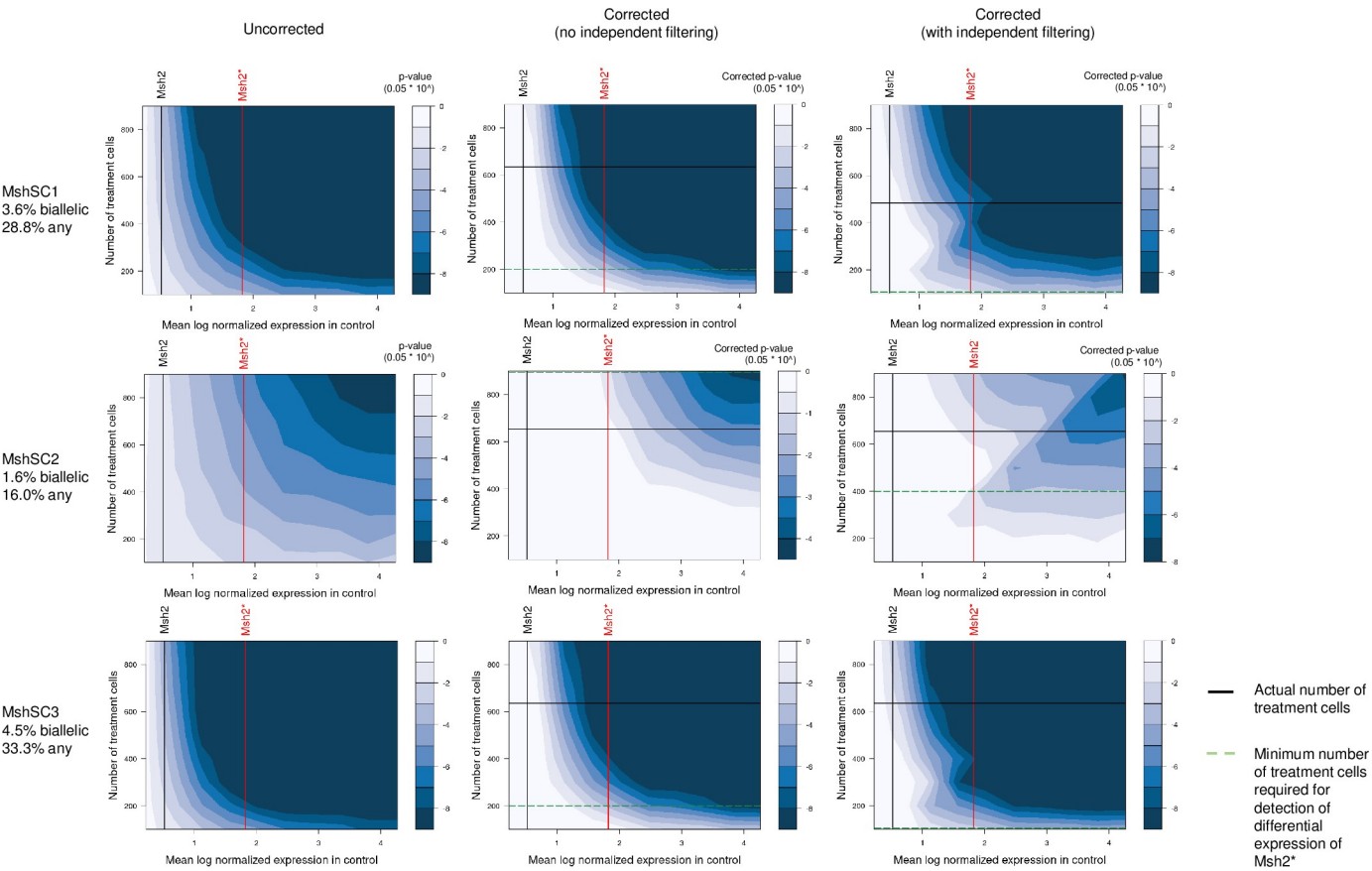

**Fig 5. Simulated-based power analysis for detecting downregulation of Msh2 with varying size of treatment population.** Contour maps depicting raw (left) and adjusted (middle, right) p-values for detecting down-regulation of target gene given fraction of monoallelic and biallelic loss for Msh2. p-values are calculated by simulating partial and full loss of genes within each gene bucket corresponding to the observed monoallelic and biallelic loss for the given number of treatment cells, and then performing differential expression via Wilcoxon rank sum. p-values are adjusted either for all genes tested (middle), or for the set of genes with baseline mean expression above the gene with the lowest baseline mean expression in that bucket, i.e. after independent filtering (right). Vertical lines indicate base expression of genes in control population with (red) and without (black) targeted sequencing. Black horizontal lines indicate the actual number of treatment cells observed, while horizontal green dashed lines indicate the minimum number of cells required to achieve significance at corrected p-value < 0.05 to detect differential expression of Msh2 with transcript-targeted sequencing.

genes with lower expression than the expression bucket the gene of interest falls in. Thus, for a multiplexed screen, the total cell count required is the sum of the required cells for the base expression of the lowest-expressing gene of interest, across gRNAs.

## Discussion

We introduced pAC-Seq, a method for the direct and robust measurement of CRISPR/Cas9 gRNAs along with transcriptomes for thousands of cells in a single experiment. Using pAC-Seq, we can assign gRNAs to >90% of cells in an scRNA-seq experiment. In addition to efficient gRNA detection, pAC-Seq's 25A-gRNA backbone allows flexibility in how gRNAs are delivered to cells. gRNAs can be directly cloned into a 25A-backbone, eliminating the time-consuming and costly process of separately sequencing each gRNA library for a correlated barcode transcript. We expect pAC-Seq will extend to experiments involving Cas9 RNP delivery using in vitro transcribed 25A-gRNAs, which expands scRNA-seq potential to hard-to-transfect primary cells [30,31]. pAC-Seq also should allow simple scRNA-seq application of pairwise gRNA screens [40]. While our experiments utilize arrayed gRNA delivery to ensure that each

cell receives a single gRNA species, our method should extend seamlessly to pooled gRNA screens, as the scRNA-seq was performed in a pooled format.

We used pAC-Seq to demonstrate that detecting changes in gene expression caused by CRISPR/Cas9 mutation of non-coding elements is challenging with contemporary scRNA-seq technology. When we used gRNAs pre-selected from MERA screens to be maximally disruptive to gene expression, ~20–30% of cells receiving cis-gRNAs are observed to exhibit some loss of target expression, while <5% of cells that receive cis-gRNAs have biallelic loss of target gene expression. This low rate of biallelic loss could be due to the relatively small size of the mutations induced by CRISPR-induced error-prone repair (<50bp) [14] with respect to the size of enhancers, which distal DNase I hypersensitive sites suggest to be ~300bp [41,42]. Furthermore, error-prone repair often results in diverse outcomes that could have varying impact on the target gene [43]. We show that pAC-Seq enables the detection of decreased expression of target genes in a pooled format, suggesting its utility in pooled screens when unbiased transcript measurement is desired. However, our resolution of phenotypic changes is limited by transcript dropout in contemporary scRNA-seq droplet-based methods [33] where initial capture of cellular RNAs is estimated to be below 10%. Instead, transcript-targeted sequencing could be used to increase power for a gene set of interest, as in Replogle et al [22].

Our results show that there is a resolution trade-off when screening the regulatory genome using single-cell technology. CRISPRi [44] and paired gRNA deletion [45] provide more defined and consistent gene expression change phenotypes by altering the regulation of hundreds to thousands of genome bases at once, and CRISPRi has been used effectively in scRNA-seq to probe combinatorial logic of enhancer function [25]. Thus, these techniques, which are equally amenable to pAC-Seq polyadenylation of their guide RNAs, should more readily scale to highly multiplexed pooled screening of non-coding phenotypes. On the other hand, we show that single gRNA mutations tend to span <50 bp, allowing us to pinpoint the causal non-coding region much more accurately [14].

Although challenges remain in understanding the phenotypic effects of germline and somatic non-coding mutations in genetic disease in cancer, there is benefit to employing methods which lie at different points on the trade-off of robustness and spatial resolution. To aid future experimental design targeting gene cis-regulatory elements, we have developed a power analysis framework based on our experimental observations. Our power analysis framework provides recommendations as to the minimum number of cells required per gRNA at the desired level of detection, and identifies if transcript-targeted sequencing is required, given estimates of the expected mono- and bi-allelic loss fractions. Our consistency evaluation of our modeling framework with experimental results is currently limited to a single replicate of our experimental design. The framework can be adapted to other settings by modifying the input mono- and bi-allelic loss fraction, or extended by implementing other perturbation outcomes such as overexpression. Future work would test the generalizability of this framework to more technical replicates, as well as more complex experimental designs.

## Materials and methods

### Plasmids, cloning, and molecular biology

For stable Cas9 expression, we constructed a Tol2 transposon site-flanked Cas9 expression plasmid with Blasticidin resistance cassette p2Tol-CAG-Cas9-BlastR, whose sequence we provide in S3 Table. The standard gRNA (wt-gRNA) plasmid, Tol2 transposon site-flanked U6 promoter-driven gRNA plasmid with Hygromycin resistance cassette p2Tol-U6-2xBbsI-sgRNA-HygR, has been reported previously [38]. To add a 25 nt poly-adenine tract to the end of the gRNA hairpin, we PCR-amplified the stretch of p2Tol-U6-2xBbsI-sgRNA-HygR

spanning from the U6 promoter until the gRNA terminator, adding a 25 nt adenine stretch prior to the terminator and cloning it into the Eco0109I-XbaI sites using Infusion (Clontech). We refer to this 25A-gRNA plasmid as p2Tol-U6-2xBbsI-gRNA-25A-HygR and provide its sequence in S3 Table.

Cloning specific gRNA sequences into the p2Tol-U6-2xBbsI-sgRNA-HygR and p2Tol-U6-2xBbsI-gRNA-25A-HygR plasmids was accomplished through pooled Gibson Assembly (NEB) using primers listed in S3 Table. gRNA sequences used in this study are listed in S1 Table where the middle region is the gRNA spacer and the flanks allow PCR amplification and Gibson Assembly.

Reverse transcription was performed with the Protoscript First Strand Synthesis Kit (NEB). Standard oligo(dT) primer was provided in this kit, and the sequencing adapter-tailed reverse transcription primer is listed in S3 Table. RT-qPCR primers are listed in S3 Table.

## Cell culture and knock-in lines

mESC culture was performed according to previously published protocols [38]. All lines were derived from a 129P2/OlaHsd background and cultured in Knockout DMEM (Thermo Fisher) with 15% defined fetal bovine serum (Thermo Fisher), 0.1 mM nonessential amino acids (Thermo Fisher), Glutamax (Thermo Fisher), 0.55 mM 2-mercaptoethanol (Sigma), ESGRO LIF (Millipore), 5 nM GSK-3 inhibitor XV (Sigma) and 500 nM UO126 (Sigma).

GFP/mCherry double knock-in mESCs were derived using the self-cloning CRISPR (scCRISPR) method [38] using primers listed in S3 Table. Flow cytometric sorting was performed using a MoFlo Astrios and analysis performed using a Cytek DXP 11.

## MERA screening and analysis

A library of gRNAs was designed according to previously published criteria [14]. The library contained a total of 12,472 gRNAs including 113 negative control gRNAs that do not target the mouse genome and 77 positive control gRNAs targeting GFP. The library was ordered from CustomArray and introduced to Msh2$^{GFP}$ mESCs using previously published techniques [14]. Two replicates of MERA screening were performed with genomic DNA collected on bulk cells and doubly flow cytometrically sorted GFP$^-$ cells. Library prep, deep sequencing, and data processing and analysis were performed using previously published protocols [14]. Deletion of candidate required regions was performed using the scCRISPR method [38], and flow cytometric analysis was used to measure the rate of GFP loss.

## Deep sequencing of genomic DNA and analysis

Deep sequencing of genomic DNA from flow cytometrically purified GFP$^+$mCherry$^+$ and doubly sorted GFP$^-$mCherry$^-$ cells was performed according to previously published protocols [38]. Genomic DNA was isolated 1–2 weeks after transfection, and PCR amplicon sequencing was performed using primers listed in S3 Table. Because indels are predominantly short (<20 nt), there is unlikely to be strong selection for deletion-bearing fragments. Sequencing adapters were extended using the NEBNext Multiplex Oligos for Illumina kit (NEB).

## Single cell RNA-sequencing and analysis

Wildtype mESCs MERA lines (control, 3 sgMsh2, 3 sgTdgf1, and 3 sgZfp42 lines) were mixed at equal concentrations on a 10 cm standard culture dish with mESC 2i media (Fig 3A). Following 48 hours, the pool was prepared for 10X Chromium scRNA-seq. Cells were single-cell dissociated with Accutase (Sigma-Aldrich), centrifuged and resuspended in 1mL pre-chilled DPBS+0.04%BSA. Cells were counted and resuspended in DPBS+0.04%BSA following the

10xChromium Volume Calculator Table for a total of 10,000 targeted cells. Similarly, Cis-targeted double negative GFP⁻mCherry⁻ cell lines (3 Msh2GFPCherry + sgMsh2 lines and 3 Tdgf1GFPCherry + sgTdgf1) and 4 GFP⁺Che⁺ cross-targeted cells (2 gRNA control and 2 sgZfp42) were mixed at equal concentrations and after 48 hours were prepared for scRNA-seq.

Preparation of 10X Chromium scRNA-seq transcriptome libraries for sequencing was carried out largely using the standard protocol. In order to maintain gRNAs in the library, which are shorter than standard transcripts, SPRI purification was altered as follows: purification of cDNA was carried out with 1.6X beads instead of the standard 0.8X, post-fragmentation purification was performed with 1.6X instead of 0.8X, post-adapter ligation purification was performed with 1.4X instead of 0.8X, and post-SI PCR was performed with 1.2X instead of 0.8X. Additionally, to reserve a portion of cDNA and allow for custom applications such as gRNA sequence capture, two additional PCR cycles (12 instead of 10) were performed at the cDNA amplification stage. Half of the product was used for the transcriptome prep, ¼ was amplified to obtain the required amount for gRNA sequence capture, and 1/8 was used for gene-specific PCR enrichment.

To enrich for gRNA transcripts in the cDNA, we performed a custom sequence capture-based library prep. An expanded version is presented in S1 File. 500ng amplified cDNA was mixed with 5ug Cot-1 DNA (Thermo Fisher) and 1nmol of each of the blocking oligos (S3 Table). The mixture was vacuum concentrated to dry pellet in a speedvac on medium for 15'-1hr. This was resuspended in 8.5ul xGen 2x Hybridization buffer and 2.7ul xGen Hybridization buffer enhancer (IDT). 1.8ul Nuclease free water was added to make up to 13ul volume. The mixture was incubated for 5' and mixed with a pipette afterwards. Content was then incubated in a thermal cycler @ 95C for 10min. Immediately after this, 4ul of 0.75pmol/ul suspension of gRNA sequence capture probe (S3 Table) was added, vortexed and briefly spun down. This was then incubated @65C for 24hrs in a thermal cycler (75C lid temperature).

Following 24 hours of incubation, the mixture was added to 75ul MYOne Streptavidin C1 (Thermo Fisher) beads washed and equilibrated in 1X bead wash buffer in a PCR tube. Components were mixed thoroughly with pipette. The DNA bead mixture was incubated in a thermal cycler at 65C for 45min with intermittent vortexing every 12 minutes. Following incubation, 65 degree and room temperature washes were performed. After the final wash, 20ul nuclease free water was added to use as a template in PCR.

Post-capture PCR (100ul) was carried out with NEBNext mix (98C – 30s; 8 cycles of 98C – 10s, 62C – 30s, 72C – 30s; 72C – 5') for 8 cycles with adapter half-tail primers (S3 Table). The PCR product was purified using 1.0X PCR volume of Ampure XP beads (Beckman Coulter, 100ul to 100ul PCR). PCR and purification were performed with the Dynabeads still in the mix. Purified DNA was eluted with 40ul Qiagen Buffer EB. Quantitative PCR was done using 0.2ul of eluted DNA to determine cycle count. Post-capture PCR 2 (100ul) was carried out to add library barcodes and full adapters using NEBNext mix (98C – 30s; X cycles of 98C – 10s, 65C – 30s, 72C – 30s; 72C – 5') using full-tail primers (S3 Table). The final PCR product was purified with Ampure XP beads and eluted with 40ul Qiagen Elution Buffer.

gRNA reads were mapped using a custom local aligner to the known set of gRNA sequences. The mapped reads were then processed with the 10X cellranger pipeline (version 1.3.1) [46]. UMI-unique counts for each gRNA were modeled as a two-component mixture model, with one component corresponding to counts arising from noise, and another component corresponding to counts arising from the presence of the gRNA in the cell. Models were initialized using the clustering results of a Gaussian mixture model, and then fit using expectation-maximization. A gRNA was defined to be present in a cell if the probability of the counts arising from the noise component was less than 0.05. Cells containing a single gRNA and that had a high quality transcriptome were retained for downstream analysis.

For comparison between existing scRNA-seq protocols that detect gRNAs, detection rate was defined as the fraction of cells with high quality transcriptomes that could be mapped to one specific gRNA. High quality transcriptomes were defined using the 10X filter as all top barcodes with UMI-unique counts within the same order of magnitude, such that the lowest quality cell should have >10% of counts of the top nth barcode, where n is 1% of the expected recovered cell count. To compare gRNA UMI counts with CROP-seq, raw data was downloaded from GEO (GSE92872) and processed using the pipeline made available by the authors (https://github.com/epigen/crop-seq, commit: 16d2bad) [20].

Transcriptome reads were mapped using the 10X cellranger pipeline to the mm10 reference genome. We defined cells retained by the 10x pipeline to have high quality transcriptomes. This cell set was used to define the detection rate for the sake of comparison. We then furthered filtered cells which were outliers for three quality control metrics: fraction of mitochondrial reads, number of gene observed, and number of UMIs. Outliers were defined as in Amezquita et al. [47] as being 3 median absolute deviations away from the median value across all cells. For analysis of differential expression of target genes and for visualization, counts were first normalized using Seurat's default pipeline [48]. To determine if genes were differentially expressed in the cell populations that received a targeting gRNA, the Wilcoxon rank sum test was used to test for a difference in median expression. Multiple testing was controlled for using Bonferroni correction with a false discovery rate of $< = 0.05$. A gene is said to be consistently differentially expressed if it is differentially expressed across the cell populations receiving a gRNA targeting the cis-regulatory regions of the same gene. Overall, results were insensitive to the differential expression method used (S10 Fig and S2 File). DESeq2 [49] and MAST [50] similarly found that target genes were differentially expressed in sorted populations and that Trh was consistently differentially expressed in sorted populations of cells receiving Msh2-targeting gRNAs. To visualize the data, principal components analysis was first run on the transformed counts to reduce the number of dimensions to 25. These were then used as input features for TSNE.

Simulation-based power analysis for detecting loss of expression was conducted as described in the main text. The framework requires as input a control cell population, and the estimated mono- and bi- allelic loss rates. Users may also pass as options the size of the gene buckets, the range over the number of treatment cells to test, or a custom differential expression method. An R package implementing the simulation framework (powerpAC) and an accompanying tutorial are provided in S3 File.

## Supporting information

**S1 Fig. Consistently robust activity of 25A-gRNAs.** Sanger sequencing analysis of nine 25A-gRNAs targeting cis-regulatory regions of Msh2, Tdgf1 and Zfp42 exhibits consistently robust mutagenic activity.
(TIF)

**S2 Fig. Deletion of Msh2 MERA hit regions induces Msh2-GFP loss.** Flow cytometric percentage of Msh2-GFP- cells after targeting with a pair of gRNAs in the GFP ORF (sgGFP- Del) or flanking each of three Msh2 regions detected as hits in the MERA screen. Enrichment of % Msh2-GFP- cells is significant for all non-control gRNAs with respect to sgControl by one-sided t-test (p-value < 0.05).
(TIF)

**S3 Fig. Cis-gRNA targeting of Tdgf1$^{GFP/mCherry}$ knock-in lines.** (A) Flow cytometry of Tdgf1$^{GFP/mCherry}$ cells shows uniformly strong bi-allelic fluorescence. (B) Tdgf1 MERA GFP-

enrichment of ~4,000 cis-gRNAs from Rajagopal et al study, highlighting locations of cis-gRNAs used in this work. (C) Flow cytometry of Tdgf1$^{GFP/mCherry}$ cells after sgTdgfcis1-3 targeting, showing robust fluorescence loss but rare bi-allelic expression loss.
(TIF)

**S4 Fig. Gene-specific enrichment increases UMI-unique counts for each transcript without introducing substantial skewing of the relative abundance.** (A) gRNA count fraction vs. log total gRNA counts in each cell in the sorted population for MshSC1, the control gRNA, as well as the most abundant gRNA in each cell (B) PCR-based enrichment of specific transcripts increases UMI-unique reads without skewing relative abundance. Normalized UMI-unique transcriptome expression (X-axis) and gene-specific expression (Y-axis) of Msh2 (left), Tdgf1 (center), and Zfp42 (right) in the sorted (top) and unsorted (bottom) experiments.
(TIF)

**S5 Fig. Flow cytometric purity of GFP⁻mCherry⁻ populations after double sorting.** (A) Flow cytometry plots showing purity of 25A-gRNA-targeted and double GFP⁻mCherry⁻ flow cytometrically purified populations. All populations are >85% pure, although single and double positive subpopulations due to imperfect sorting and/or re-expression of transgenes after sorting. (B) RT-qPCR expression of Msh2 (left plot) and Tdgf1 (right plot) in control gRNA-targeted (left), bulk cis-gRNA targeted (middle), and GFP⁻mCherry⁻ double-sorted (right) populations, showing strong flow cytometric enrichment of cells lacking target gene expression.
(TIF)

**S6 Fig. Cis-gRNA targeting of Zfp42$^{GFP}$ knock-in line.** (A) Zfp42 MERA GFP⁻ enrichment of ~4,000 cis-gRNAs from Rajagopal et al study, highlighting locations of cis-gRNAs used in this work. (B) Flow cytometry of Zfp42$^{GFP}$ cells after sgZfp42cis1-3 targeting, showing robust fluorescence loss.
(TIF)

**S7 Fig. Limits of scRNA-seq resolution in analysis of wildtype experiment.** (A) t-SNE plot of wildtype pAC-Seq experiment labeled by gRNA species. t-SNE plot of the wildtype experiment showing cells colored by the gRNA they received. There is no clear separation of cells based on gRNA expression, as would be expected given that cis-gRNAs have subtle effects on the transcriptome. (B) Distribution of log-normalized expression of Msh2 and Tdgf1 in cells receiving Msh2-targeting gRNAs (above) and Tdgf1-targeting gRNAs (below) with transcript-targeted sequencing in wildtype experiment.
(TIF)

**S8 Fig. Simulated-based power analysis for detecting downregulation of Tdgf1 with varying size of treatment populations.** Contour maps depicting raw (left) and adjusted (middle, right) p-values for detecting down-regulation of target gene given fraction of monoallelic and biallelic loss for Tdgf1. p-values are calculated by simulating partial and full loss of genes within each gene bucket corresponding to the observed monoallelic and biallelic loss for the given number of treatment cells, and then performing differential expression via Wilcoxon rank sum. p-values are adjusted either for all genes tested (middle), or for the set of genes with baseline mean expression above the gene with the lowest baseline mean expression in that bucket, i.e. after independent filtering (right). Vertical lines indicate base expression of genes in control population with (red) and without (black) targeted sequencing. Black horizontal lines indicate the actual number of treatment cells observed, while horizontal green dashed lines indicate the minimum number of cells required to achieve significance at corrected p-

value < 0.05 to detect differential expression of Tdgf1 with transcript-targeted sequencing.
(TIF)

**S9 Fig. Overlap of differentially expressed genes across Msh-targeting gRNAs and differential expression methods vs. ZfpSC1.** (A) Overlap across different differential expression methods for each Msh2-targeting gRNA in the sorted population. (B) Overlap across Msh-targeting gRNAs for each differential expression method in the sorted population. (C) Overlap across different differential expression methods for consistently differentially expressed genes identified across Msh2-targeting gRNAs in the sorted population. (A-C) Numbers in parenthesis indicate the number of differentially expressed genes identified at adjusted p-value < 0.05 (D) Trh was found to be differentially expressed for MshSC3 across all methods, and for MshSC1 using DESeq2 (indicated by *, adjusted p-value < 0.05) in the unsorted population.
(TIF)

**S10 Fig. Simulation-based power analysis for detecting downregulation of target-gene using MAST with varying size of treatment populations.** Contour maps depicting raw (left) and adjusted (middle, right) p-values for detecting down-regulation of target gene given fraction of monoallelic and biallelic loss for (A) Msh2 and (B) Tdgf1. p-values are calculated by simulating partial and full loss of genes within each gene bucket corresponding to the observed monoallelic and biallelic loss for the given number of treatment cells, and then performing differential expression via MAST. p-values are adjusted either for all genes tested (middle), or for the set of genes with baseline mean expression above the gene with the lowest baseline mean expression in that bucket, i.e. after independent filtering (right). Vertical lines indicate base expression of genes in control population with (red) and without (black) targeted sequencing. Black horizontal lines indicate the actual number of treatment cells observed, while horizontal green dashed lines indicate the minimum number of cells required to achieve significance at corrected p-value < 0.05 to detect differential expression with transcript-targeted sequencing.
(TIF)

**S1 Table. Results from MERA screen targeting regulatory regions of Msh2.**
(PDF)

**S2 Table. Summary of gRNA counts from the original CROP-seq publication vs. pAC-seq.**
(PDF)

**S3 Table. Table of oligonucleotides.**
(PDF)

**S1 File. Detailed sequence capture protocol for gRNAs.**
(DOCX)

**S2 File. Comparison of differential expression results across different differential expression methods.**
(ZIP)

**S3 File. R package (powerpAC) and accompanying tutorial for power analysis simulations.**
(ZIP)

## Acknowledgments

The authors thank Yiling Qiu and Chad Araneo for technical assistance.

## Author Contributions

**Conceptualization:** Grace Hui Ting Yeo, Richard I. Sherwood, David K. Gifford.

**Formal analysis:** Grace Hui Ting Yeo, Oscar Juez, Qing Chen, Budhaditya Banerjee, Lendy Chu, Max W. Shen, May Sabry, Ive Logister, Richard I. Sherwood, David K. Gifford.

**Funding acquisition:** Richard I. Sherwood, David K. Gifford.

**Investigation:** Oscar Juez, Qing Chen, Budhaditya Banerjee, Lendy Chu, May Sabry, Ive Logister, Richard I. Sherwood, David K. Gifford.

**Methodology:** Grace Hui Ting Yeo, Oscar Juez, Qing Chen, Budhaditya Banerjee, Lendy Chu, May Sabry, Ive Logister, Richard I. Sherwood, David K. Gifford.

**Project administration:** Richard I. Sherwood, David K. Gifford.

**Resources:** Richard I. Sherwood, David K. Gifford.

**Software:** Grace Hui Ting Yeo.

**Supervision:** Richard I. Sherwood, David K. Gifford.

**Validation:** Oscar Juez, Qing Chen, Budhaditya Banerjee, Lendy Chu, May Sabry, Ive Logister, Richard I. Sherwood.

**Visualization:** Grace Hui Ting Yeo, Richard I. Sherwood.

**Writing – original draft:** Grace Hui Ting Yeo, Richard I. Sherwood, David K. Gifford.

**Writing – review & editing:** Grace Hui Ting Yeo, Richard I. Sherwood, David K. Gifford.

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
