## [Decision Letter · Decision Letter 0]

24 Sep 2020

Dear Dr. Gifford,

Thank you very much for submitting your manuscript "Detection of gene cis-regulatory element perturbations in single-cell transcriptomes" for consideration at PLOS Computational Biology.

As with all papers reviewed by the journal, your manuscript was reviewed by members of the editorial board and by several independent reviewers. In light of the reviews (below this email), we would like to invite the resubmission of a significantly-revised version that takes into account the reviewers' comments.

All reviewers agree that the experimental method presented is important and advances the field.  As you prepare your revision, I would encourage you to focus in particular on reviewer comments related to computational analysis and modeling, given the audience of PLOS Computational Biology.

We cannot make any decision about publication until we have seen the revised manuscript and your response to the reviewers' comments. Your revised manuscript is also likely to be sent to reviewers for further evaluation.

Sincerely,

Christina S. Leslie

Associate Editor

PLOS Computational Biology

Weixiong Zhang

Deputy Editor

PLOS Computational Biology

Reviewer's Responses to Questions

**Comments to the Authors:**

Reviewer #1: In this manuscript, Yeo et al present a novel design of sgRNAs that enables direct detection of sgRNAs during single-cell RNA-seq. The authors accomplish this by appending a polyA tail downstream of the sgRNA sequence. They then identify non-coding regions to test using their CRISPR screening approach, and test pAC-Seq on these regions.

Overall, the general concept of this paper is clear and has potential to improve recently developed single-cell perturbation assays. I reviewed a previous version of this manuscript at a different journal over 2 years ago and my previous concerns have been addressed in this new version.

Reviewer #2: The manuscript by Yeo and collaborators presents a new protocol to directly measure gRNAs in scRNA-seq. Poly-adenine CRISPR gRNA-based scRNA-seq (pAC-Seq) is proposed as a method to edit regulatory elements and measuring the effect effect on gene expression through scRNA-seq. The method is very relevant for the field as it combines the ability to directly measuring gRNA in each individual cell without the limitations of barcode swapping due to lentiviral recombination. Despite the excitement for the potential technological advancement associated with this new method, several limitations were identified that overall diminish this reviewer’s enthusiasm, as detailed below.

Major comments:

1. It appears that the results presented are from individual experiments of pAC-Seq, without experimental replicates. While in the very first studies of scRNA-seq it was acceptable to consider individual cells within one experiment as technical replicates, this is not the case anymore. Technical replicates are especially important in this case to demonstrate that the results of the editing experiment are consistent across technical iterations of the protocol.

2. Minimal consideration is given to off target effects, by specifically analyzing only the genes of interest. Even in the only analysis where widespread effects on gene expression are considered, the use of the MWU test is not appropriate. A negative binomial model that directly models count data (DESEQ2) or another method specifically suited for scRNA-seq would be more appropriate for the analysis of differential gene expression in these experiments.

3. There are very limited details on the modeling analysis to aid in future study design. Most importantly, no experimental validation is provided on whether the outcome of the modeling is actually predictive of the best parameters for designing future studies.

4. The manuscript claims that this method has high multiplexing potential and could be used to study a large number of regulatory elements, however this potential is not demonstrated in the manuscript, rather the method seems to have limited sensitivity and requires a relatively large number of cells to detect allelic events, which may not be amenable to high multiplexing studies.

Minor comments.

Introduction: The first paragraph of the introduction should reference MPRA/STARR-seq studies that have provided answers to some of the specific questions highlighted: identify functional regulatory elements and their regulatory effects on gene expression, analyze the function of non-coding mutations.

Figure 1B: In the caption please explain what the horizontal segment indicates.

Line 149: How is the significant enrichment defined?

Line 290: The manuscript states that “the data was insufficient to impute with confidence” allelic expression. Please specify why the data were insufficient. Would a different study design (e.g. number of cells analyzed) be more appropriate to ask this question. As this was one of the major goals of this new method, new experiments to address this limitation should be considered.

Reviewer #3: Overall:

Yeo et al present highly creative, thorough, and well-written work. They successfully demonstrate a novel gRNA-capture single-cell RNA-seq method with high editing efficiency and high single-cell capture rates. Furthermore, they pursue a sequence-interruption based perturbation of the noncoding genome, a risky yet required step in the field’s application of single-cell perturbation screens to study noncoding elements. Overall, the work is thorough, detailed, and advances the field. However, the manuscript lacks some discussion of 1) why such they detect such a low rate of gene expression reduction, and 2) the pros/cons of targeted capture single-cell RNA seq in these screens. I recommend acceptance into PloS Computational Biology with these two additional discussions and other minor changes.

Major

1) The authors cite lack of signal according to single-cell RNA’s lack of power to detect [Lines 171-174; Fig S1; Line 388-389.]. However, though it is implied, I think this should be clearly stated that this is a) likely due to the very small effect sizes that noncoding sequence changes of < 50 basepairs will have upon enhancers/promoters (this could be phrased in citations from the MPRA field, the length of TFBSs, etc) and b) the diverse indel editing outcomes from one gRNA adds an extreme amount of noise to their assay. Even if a small sequence change has an effect (as in A) it is very unlikely it would be detectable given the inability to genotype the sequence change per cell (as in B). Please add discussion clarifying this, and why it could underlie the detected <5% loss of target gene expression.

2) As mentioned in lines 240-243, the authors are performing a targeted single-cell RNA seq experiment. Please clarify this earlier in the work (ideally the abstract): especially now that 10x Genomics offers custom targeted single-cell RNA-seq panels (https://www.10xgenomics.com/products/targeted-gene-expression). I applaud the authors for including it in their Fig 5 simulation, as this will be very useful for future users.

Minor

Lines 57: Xie et al (Mosaic-seq) should be cited in addition to Gasperini et al. Additionally, another flaw of CRISPRi is it is unknown whether the KRAB-repression accurately perturbs enhancer/promoter function.

Figure 1A: For clarity, please include a clearer diagram of the construct, ideally with actual letters ie a portion of the U6 promoter, ~20Ns, the actual hairpin sequence, and the 25xAs, and the Pol3 termination sequence.

Use of “protospacer” vs “spacer”: If I am not mistaken, the ‘protospacer’ sequence refers to the in-genome CRISPR-target DNA sequence (hence its use of the term ‘protospacer adjacent motif’). The “spacer” is the sequence included on the actual guide-RNA itself. Please verify and correct this use in the manuscript ie “We then cloned three GFP-targeting protospacers into either the wildtype gRNA “ changed to “three GFP-targeting spacers”. (Less an issue with the manuscript and more with confusion in the field itself).

Line 80: Polyadenine spelling typo.

Fig 1: This is a higher editing rate than I would expect. Were the GFP gRNAs used in Fig1 newly designed or previously published? Please clarify either gRNA design or cite. Additionally please clarify in the Materials & Methods a) the protocol used to Sanger edits (was PCR used? Due to length bias, this can enrich for deletion-bearing fragments depending on the protocol) and b) the length of time between gRNA delivery and phenotyping.

Figure 3B: Please provide a similar figure showing plotting not just an individual gRNA but of any gRNAs in the population (ie plot each cell by its most-abundant gRNA, and do not limit which gRNA it might be on the graph). Additionally, please include a brief discussion for why the gRNA count-fraction is < 0.5, rather than closer to 1 given there should only be 1 gRNA per cell.

Figure 3C: Perhaps I am missing it, but please include the number of cells that do not have any gRNA assigned to them. This will clarify the % assignment rates in Line 229 (75.3% and 91.2%) for each experiment. Why do these assignment rates differ between the experiments?

Line 481: The sequence capture method listed here is complicated. In comparison, did you attempt a simpler PCR-enrichment in parallel (as performed in citation 22)? Which would you recommend for pAC-seq?

Figure 5: This simulation is a highly useful tool for the field, and I commend the authors for including it.

**Have all data underlying the figures and results presented in the manuscript been provided?**

Reviewer #1: Yes

Reviewer #2: Yes

Reviewer #3: Yes

PLOS authors have the option to publish the peer review history of their article (what does this mean?). If published, this will include your full peer review and any attached files.

Reviewer #1: No

Reviewer #2: No

Reviewer #3: No
---

## [Decision Letter · Decision Letter 1]

13 Feb 2021

Dear Dr. Gifford,

We are pleased to inform you that your manuscript 'Detection of gene cis-regulatory element perturbations in single-cell transcriptomes' has been provisionally accepted for publication in PLOS Computational Biology.

Additionally, you will note that one reviewer still had significant concerns that the main pAC-Seq experiment was presented without a technical replicate and that the simulation framework was calibrated on this experiment without demonstrating generalization to other experiments. While other reviewers did not share these concerns, it would be appropriate to add a short paragraph to the Discussion to briefly outline limitations of the current study. I am copying the Deputy Editor here so that the editorial team is aware of this minor requested addition.

Best regards,

Christina S. Leslie

Associate Editor

PLOS Computational Biology

Weixiong Zhang

Deputy Editor

PLOS Computational Biology

Reviewer's Responses to Questions

**Comments to the Authors:**

Reviewer #1: In this revision, the authors have addressed all of my previous concerns.

Reviewer #2: This revised manuscript addresses some of the comments raised during the first round of review, but two main comments remain unaddressed, as detailed below.

The first one is about presenting clear data on the reproducibility of the experimental method. As with every new experimental protocol, assessing reproducibility is key and essential. This is different than presenting validation data using independent methods, which the manuscript includes and are of course of value. However, it is important for future users to know what is the reproducibility of the method itself. If the same experiment is repeated twice, how reproducible are the results?

The second issue is with the simulation framework. The simulations are calibrated based on the observed experimental data (e.g. down-regulation of gene expression) and then evaluated against the results of the same experiments. Obviously, the simulations are found to well represent the observed experimental data, but this is expected given that the same data are used to calibrate them. Thus, the manuscript does not present convincing evidence that the proposed simulation framework would work for an independent dataset/experiment. Independent statistical or experimental validation of the simulation framework should be presented.

Reviewer #3: The authors have met all my concerns.

**Have all data underlying the figures and results presented in the manuscript been provided?**

Reviewer #1: None

Reviewer #2: Yes

Reviewer #3: None

PLOS authors have the option to publish the peer review history of their article (what does this mean?). If published, this will include your full peer review and any attached files.

Reviewer #1: No

Reviewer #2: No

Reviewer #3: No

---

## [Editor Report · Acceptance letter]

8 Mar 2021

PCOMPBIOL-D-20-01385R1 

Detection of gene cis-regulatory element perturbations in single-cell transcriptomes

Dear Dr Gifford,

I am pleased to inform you that your manuscript has been formally accepted for publication in PLOS Computational Biology. Your manuscript is now with our production department and you will be notified of the publication date in due course.

With kind regards,

Alice Ellingham
